# Predictors of Phthalate Metabolites Exposure among Healthy Pregnant Women in the United States, 2010–2015

**DOI:** 10.3390/ijerph20237104

**Published:** 2023-11-23

**Authors:** Shabnaz Siddiq, Autumn M. Clemons, John D. Meeker, Chris Gennings, Virginia Rauh, Susannah Hopkins Leisher, Adana A. M. Llanos, Jasmine A. McDonald, Blair J. Wylie, Pam Factor-Litvak

**Affiliations:** 1Department of Epidemiology, Mailman School of Public Health, Columbia University, New York, NY 10032, USA; ama2217@cumc.columbia.edu (A.M.C.); al4248@cumc.columbia.edu (A.A.M.L.); prf1@cumc.columbia.edu (P.F.-L.); 2Department of Environmental Health Sciences, University of Michigan School of Public Health, Ann Arbor, MI 48109, USA; meekerj@umich.edu; 3Department of Environmental Medicine and Public Health, Icahn School of Medicine at Mount Sinai, New York, NY 10029, USA; chris.gennings@mssm.edu; 4Heilbrunn Department of Population and Family Health, Mailman School of Public Health, Columbia University, New York, NY 10032, USA; var1@cumc.columbia.edu; 5Stillbirth Research Program, Department of Obstetrics & Gynecology, University of Utah School of Medicine, Salt Lake City, UT 84112, USA; 6Department of Obstetrics and Gynecology, Columbia University Vagelos College of Physicians and Surgeons, New York, NY 10032, USA; ebj2107@cumc.columbia.edu

**Keywords:** phthalate metabolites, pregnancy, predictors, urine, environmental epidemiology

## Abstract

Phthalate use and the concentrations of their metabolites in humans vary by geographic region, race, ethnicity, sex, product use and other factors. Exposure during pregnancy may be associated with detrimental reproductive and developmental outcomes. No studies have evaluated the predictors of exposure to a wide range of phthalate metabolites in a large, diverse population. We examined the determinants of phthalate metabolites in a cohort of racially/ethnically diverse nulliparous pregnant women. We report on urinary metabolites of nine parent phthalates or replacement compounds—Butyl benzyl phthalate (BBzP), Diisobutyl phthalate (DiBP), Diethyl phthalate (DEP), Diisononyl phthalate (DiNP), D-n-octyl phthalate (DnOP), Di-2-ethylhexyl terephthalate (DEHTP), Di-n/i-butyl phthalate (DnBP), Di-isononyl phthalate (DiNP) and Di-(2-ethylhexyl) phthalate (DEHP) from urine collected up to three times from 953 women enrolled in the Nulliparous Mothers To Be Study. Phthalate metabolites were adjusted for specific gravity. Generalized estimating equations (GEEs) were used to identify the predictors of each metabolite. Overall predictors include age, race and ethnicity, education, BMI and clinical site of care. Women who were Non-Hispanic Black, Hispanic or Asian, obese or had lower levels of education had higher concentrations of selected metabolites. These findings indicate exposure patterns that require policies to reduce exposure in specific subgroups.

## 1. Introduction

Exposure to environmental chemicals affects the health outcomes of pregnant women and their offspring. Phthalates are a class of non-persistent endocrine-disrupting chemicals (EDCs) that are widely used in plastics to increase their flexibility, transparency, durability and longevity [1]. Each year, an estimated 213 million kilograms of phthalates are produced or imported in the United States and a variety of industrial and consumer products such as polyvinyl chloride products, medical devices, food packaging, toys and personal care products contain phthalates [1,2]. Phthalates are often added to cosmetics and other personal care products as preservatives and to maintain the fragrance in cosmetics and improve the texture and consistency of some products [3,4].

Because of phthalates’ weak covalent bonds to materials, they leach or are aerosolized into the environment; the most common routes of exposure in humans are ingestion, inhalation, dermal absorption and parenteral administration, often from the plastic tubing used for medical procedures [2]. For example, exposure to Di-(2-ethylhexyl) phthalate (DEHP), a high-molecular-weight phthalate (HMWP), commonly occurs due to consumption of food and water that are stored in food packaging products and bottles containing DEHP, whereas exposure to most low-molecular-weight phthalates (LMWPs) commonly occurs via dermal exposure to personal care and cosmetic products [5]. After exposure, phthalates rapidly metabolize, and the phthalate metabolites (PthMs) are excreted in urine (main excretion route). Emerging evidence suggests that exposure to phthalates can harm human health, especially during critical periods in the life course, one being the perinatal period. Higher urinary PthM concentrations are related to adverse outcomes both in pregnant women (e.g., pre-eclampsia, gestational diabetes) and their children (e.g., preterm birth, spontaneous abortion, effect on IQ, motor function and behavior outcomes) [6,7,8,9,10,11]. Thus, to develop primary prevention programs and to understand the potential structural determinants of high exposure, it is imperative to characterize the predictors of PthMs that may impact pregnant women and their offspring during this heightened window of susceptibility.

PthMs are non-persistent compounds with short half-lives that mostly reflect exposure in the past 3 to 24 h, so reliance on a single urine sample, often from cross-sectional studies, may not represent exposure throughout pregnancy [1]. Further, the toxicologically relevant pregnancy exposure window for phthalate exposure is unknown and may differ for different outcomes [6,11,12,13]. Studies that assessed interclass correlations (ICCs) from pregnant women based on one spot and first morning urine sample collected across a week to several months found weak to moderate reproducibility (<0.4 to 0.75), with higher ICCs (i.e., greater temporal stability) for metabolites from sources of exposure that are most consistent throughout the day (such as personal care products or time spent in environments that contain phthalates in the built environment (e.g., flooring materials)) [14]. For most PthMs, therefore, a single measure may not reflect long-term exposure patterns or exposure during a “critical period”, and studies that assess long-term repeated exposure for the same women may provide a more accurate representation of average concentrations throughout pregnancy. PthMs vary by sociodemographic characteristics (e.g., race and ethnicity, socioeconomic position, geography) and lifestyle behaviors (e.g., food consumption, personal care product use) [15,16,17]. Differential product availability and use patterns, resulting in differences in urinary PthM concentrations, are likely associated with characteristics such as race and ethnicity, sex and geographic locations. Identifying predictors of phthalate use during this critical period may help tailor interventions to address high-risk populations to reduce or eliminate their exposure levels. It is also important to evaluate a range of PthMs and identify what the distribution and differences in predictors may (or may not) be among a large, diverse population as this perspective is not well understood in the literature. Further, less is known about the effects of newer phthalate replacement compounds and their effect on human health [11,18,19]. Here, we use longitudinal data from eight clinical centers across the United States (US) to investigate predictors of a wide range of PthM concentrations (including newer replacement phthalates) throughout pregnancy among a large, diverse cohort of healthy nulliparous individuals with singleton pregnancies.

## 2. Materials and Methods

### 2.1. Study Design

Nulliparous Mothers to Be Study (nuMoM2b) is a prospective cohort study that enrolled pregnant nulliparous women between 2010 and 2015 to evaluate the associations between maternal characteristics and adverse pregnancy outcomes. The methods for the study have been published previously [20]. The study personnel recruited women from eight clinical centers across the US: Case Western University; Columbia University; Indiana University; the University of Pittsburgh; Northwestern University; the University of California at Irvine; the University of Pennsylvania; and the University of Utah. Eligible women were those without previous delivery, at 20 weeks gestation or later and who had a viable singleton pregnancy with an estimated gestational age at recruitment from 6 weeks 0 days to 13 weeks 6 days. Only women who intended to deliver at one of the participating clinical hospital sites were eligible. Women with previous enrollment, <13 years of age, with a history of three or more pregnancy losses, with a donor oocyte pregnancy, with planned pregnancy termination or with potentially lethal malformation or who were unable to provide informed consent were excluded from the study. Enrolled women participated in at least three study visits (6 week 0 days to 13 weeks 6 days; 16 weeks 0 days to 21 weeks 6 days; 22 weeks 0 days to 29 weeks 6 days), one in each trimester of pregnancy. During the study visits, the study personnel administered structured questionnaires to collect data on demographic characteristics, medical history, lifestyle behaviors and other characteristics. The enrolled women also provided urine samples at each trimester of pregnancy. Additional details regarding the recruitment of women and their follow-up are described elsewhere [20].

We used data from a nested, matched case–control study within nuMoM2b designed to evaluate associations between phthalate exposure and preeclampsia, new onset antenatal hypertension and spontaneous preterm birth. For this analysis, we only included control women (*n* = 953) who did not have any of these adverse pregnancy outcomes under study and who also did not have gestational diabetes. We excluded women with adverse outcomes and gestational diabetes because exposure patterns and the risk profile are different between cases and controls.

### 2.2. Ethics Statement

The ethics committees of each clinical center listed above approved the study and all participants provided written informed consent.

### 2.3. Urinary Concentration of Phthalate Metabolites

A total of 2728 maternal urine samples were analyzed. There were 854 women who provided urine samples in all three trimesters of pregnancy; 67 provided samples in two trimesters only and 32 provided one urine sample. Seven women did not provide urine samples and thus we do not have phthalate metabolite data for them. The concentrations of phthalate metabolites in urine samples were analyzed at the NIEHS-funded Children’s Health Exposure Assessment Resource (CHEAR) Laboratory Hub located at the University of Michigan. Detailed laboratory methods for the measurement of metabolites are reported elsewhere [21,22]. Briefly, the spot urine samples were collected in polypropylene phthalate-free tubes and were pre-concentrated using online solid phase extraction (SPE). The analytes were separated using ultra-high-performance liquid chromatography (UHPLC) and then detected using heated electrospray ionization (HESI)-tandem mass spectrometry (MS/MS). The method was developed to replicate the Centers for Disease Control and Prevention (CDC) Urine Method No 6306.03. We studied the urinary metabolites of nine parent phthalates or replacement compounds: Butyl benzyl phthalate (BBzP), Diisobutyl phthalate (DiBP), Diethyl phthalate (DEP), Diisononyl phthalate (DiNP), D-*n*-octyl phthalate (DnOP), Di-2-ethylhexyl terephthalate (DEHTP), Di-*n*/i-butyl phthalate (DnBP) and ∑DEHP (see Table 1). One classification of phthalates is based on their molecular weight, which reflects structural similarity and similarity in routes of exposure. We included four low-molecular-weight phthalates (LMWPs) (<250 g/mol): DiBP, DEP, DnBP and DiNP. These are used in personal hygiene and cosmetic products [3]. We included five high-molecular-weight phthalates (HMWP) (≥250 g/mol): DEHP, BBzP, DiNP, DNOP and DEHTP. These are used in plastic tubing, toys, home products and food packaging [1,5]. Replacement compounds such as Di-2-ethylhexyl terephthalate (DEHTP) and 1,2-Cyclohexane dicarboxylic acid diisononyl ester (DINCH) are relatively new plasticizers; however, studies on the toxicity and health implications of chronic exposure to these compounds are limited [23,24,25]. The summary measure of DEHP metabolites is defined as the molar sum of mono-2-ethylhexyl phthalate (MEHP), mono-2-ethylhexyl phthalate (MEHP), mono-2-ethyl-5-hydroxyhexyl phthalate (MEHHP) and mono-2-ethyl-5-carboxypentyl phthalate (MECPP) metabolites. We detected varied levels of metabolites in our sample, ranging from 5% to 100% of the samples. Metabolites with detectable levels <70% were removed from the analysis (i.e., Mono(cis-hydroxy-isononyl) ester (MHNCH), Monocarboxyisooctyl ester (MCOCH) and Monocarboxyisononyl (MCINP)). For other PthMs, concentrations below the limit of detection (LOD) were replaced with LOD/√2 (see Table 1). Urinary dilution was corrected using the specific gravity (SG) adjustment method [6], which is preferred for pregnant women [17].

### 2.4. Assessment of Predictor Variables

We considered a wide variety of possible predictors collected as part of the larger nuMoM2b study. Trimester information was collected from enrolled women at each study visit. Women self-reported their race/ethnicity as non-Hispanic White, non-Hispanic Black/African American, Hispanic, Asian, American Indian, Native Hawaiian, other or multiracial. The latter four categories were considered together since the number of women in each group was less than 10. Educational attainment was categorized as high school (HS) or less (less than HS or HS graduate or Graduate Equivalency Degree (GED) completed), some college (some college credit but no degree or associate (assoc)/technical (tech) degree), college graduate (bachelor’s degree) or graduate degree (master’s, doctorate or professional degree). Their body mass index (BMI) was measured at the initial study visit, where women wore light clothes without shoes; their weight was measured using an electronic or balance scale and their height was measured using a stadiometer or measuring tape [20]. We categorized early pregnancy (i.e., within 6 weeks 0 days to 13 weeks 6 days) BMI as underweight (<18.5 kg/m^2^), normal weight (18.5–<25 kg/m^2^), overweight (25.0–<30 kg/m^2^) and obese (≥30 kg/m^2^) [24]. Lastly, we considered geographic locations specific to the parent study center as phthalate concentrations may differ. We did not consider smoking and drinking status since the majority of the women (>95%) did not report smoking or alcohol consumption.

### 2.5. Statistical Analysis

To assess the predictors of PthMs among healthy nulliparous women with singleton pregnancies, we first examined the distribution of sociodemographic variables in 953 participants (means, standard deviations (SD) and percentages), and the relationship of each predictor to each PthM. Since the concentrations of PthMs were positively skewed, we calculated the geometric mean (GM) of each metabolite for each trimester; this represents the average exposure levels for each trimester. We used generalized estimating equation (GEE) models to estimate the effect of each trimester on each metabolite and assess the relationships between each predictor and the urinary PthMs. GEE models make no distributional assumptions and require three specifications: a mean function, a variance function and a correlation structure [26,27]. This modeling approach estimates the average independent associations accounting for within-group correlations of concentrations of each PthM by trimester. Each GEE model included all predictors. We examined the goodness-of-fit (QIC) statistics of all phthalate metabolites to find an acceptable working correlation structure. All correlation structures examined (exchangeable, independent and autoregressive 1 correlation) provided similar QIC values, and we selected an exchangeable correlation structure. We also assessed between- and within-metabolite variability (i.e., temporal reliability) by calculating the ICC of the PthMs over pregnancy. ICCs here present the variance explained by trimester for each PthM. An ICC ≥ 0.75 represents excellent reproducibility, 0.4 to 0.75 represents fair to good reproducibility and <0.4 represents poor reproducibility [28].

In a secondary analysis, we compared women who were highly exposed to those not highly exposed. High exposure was defined as concentrations ≥90th percentile for four or more metabolites. These women will also have at least one PthM in any trimester with concentration levels ≥90th percentile. We used logistic regression to assess differences in the predictors between women highly exposed and those not highly exposed. In a supplementary analysis, we fitted generalized linear models predicting the geometric mean (over all trimesters) for each PthM. A *p*-value of < 0.05 was considered statistically significant in all multivariate models. All analyses were conducted using the SAS statistical software (version 9.4, SAS Institute INC., Cary, NC, USA).

## 3. Results

Table 2 shows the sociodemographic characteristics of the study population. Most women were between 20 and 34 years old (mean age: 26.9 years, SD: 5.8), had completed college and/or higher education (51%) and were within the normal weight range (52%). Approximately 58% of the women were non-Hispanic White and 39% of the women represented non-Hispanic Black, Hispanic and Asian races. In addition, 43% of the women were from clinical sites in the Northeast region, followed by 30% of women from the Midwest region, 20% from the Mountain States and 7% from the Western region.

Table 3 presents the distribution of phthalate metabolites by trimester. The concentrations of PthMs varied throughout pregnancy, with MBzP, MEP, MiBP, MINP and MECPTP increasing and ∑DEHP, MCOP, MCPP, MECPP and MEHHP decreasing in the second trimester. The MEHHTP levels did not change throughout pregnancy. However, these changes were not statistically significant (*p*-value < 0.05). The ICCs were low to moderate in our sample and demonstrated poor reproducibility over time (i.e., ICC = 0.001–0.11) except for MBZP (ICC = 0.52) and MIBP (ICC = 0.53) (Table 4).

Table 5, Table 6 and Table 7 present multivariate models indicating the predictors of urinary phthalate metabolite concentrations.

### 3.1. HMWP

The determinants of HMWPs (i.e., ∑DEHP, MBzP, MCOP, MCPP) among this sample of women include age, race and ethnicity, education and clinical site of care. Women aged 20–24 years old had increased ∑DEHP levels likely due to increased MECPP and MEHP levels in comparison to women aged < 20 years of age. Asian women and women who reported another race (i.e., American Indian, Native Hawaiian, other or more than 1 race) had lower concentrations of MCPP in comparison to Non-Hispanic White women on average. In addition, lower urinary ∑DEHP concentrations were found among women who had completed high school or a GED in comparison to women who had not completed high school (*p* < 0.05). Women who completed college also had lower levels of ∑DEHP in comparison to women who had not completed high school (*p* > 0.05) On average, women who were overweight or obese had significantly higher levels of MCOP levels in comparison to women of normal weight. Obese women in general had higher concentrations of all HMWPs. Lastly, women who received care in clinical sites in Philadelphia, Illinois, California and Utah had lower MBzP concentration levels and women who received care in New York City, Philadelphia and Utah had higher ∑DEHP levels and MCOP levels in comparison to women who received care in Ohio, on average (refer to Table 5).

### 3.2. LMWP

The determinants of LMWPs (i.e., MiBP, MEP and MnBP) among this sample of women include race and ethnicity, education, BMI and clinical site of care. For example, Non-Hispanic Black/African American and Hispanic women had higher levels of MiBP and Hispanic women had higher levels of MnBP compared to Non-Hispanic White women. Women with higher education attainment (i.e., who had completed degree work beyond college) had lower levels of MnBP and MEP compared to women who had not completed high school. In addition, women with a BMI < 18.5 kg/m^2^ had lower levels of MiBP compared to women who were normal weight, on average. Although not always statistically significant, underweight women generally had a lower LMWP concentration than others. The phthalate metabolite concentration levels varied across clinical sites of care. Women who received care in Chicago and California had lower levels of MnBP whereas women who received care in New York City, Illinois and California had higher levels of MEP in comparison to women who received care in Ohio (refer to Table 6).

### 3.3. Replacement Phthalates

Determinants of replacement phthalates (i.e., MECPTP and MEHHTP) among this sample of women include race, education and clinical site. Non-Hispanic Black/African American and Hispanic women had higher concentrations of MECPTP in comparison to Non-Hispanic White women, and women who had completed college or degree work beyond college had a higher urinary MEHHTP concentration in comparison to women who had not completed high school. On average, women who received care in Utah had higher levels of MECPTP and MEHHTP levels in comparison to women who received care at Ohio (refer to Table 7).

### 3.4. Additional Analyses

In a secondary analysis, we identified 826 women who had at least one PthM in any trimester with concentration levels ≥90th percentile. Among these women, approximately 30% of women were exposed to high levels of MCPP, 24% were exposed to high levels of MECPP, 17% were exposed to high levels of MBZP and the remainder were exposed to high levels of DEHP, MCOP, MECPTP, MEHHP, MEHHTP, MEHP, MEOHP, MEP, MHNCH, MIBP or MINP. We also identified predictors of PthMs for women exposed to high levels of ≥4 metabolites versus <4 metabolites (Table 8). Significant predictors were a higher early pregnancy BMI and the clinical site of care. Regarding BMI, women with a BMI ≥30 kg/m^2^ were almost twice as likely as those with a BMI 18.5–<25.0 kg/m^2^ to be exposed to ≥4 high PthMs. Women who received care at the University of Pennsylvania and the University of Utah were almost three times as likely as women who received care at Case Western Reserve University to be exposed to ≥4 high PthMs (refer to Table 8).

The results from the supplemental analysis (Table A1, Table A2, Table A3, Table A4 and Table A5, in Appendix A) showed that among the HMWPs, BMI and clinical site of care were significant predictors for MBzP and MCOP, whereas only the clinical site of care was significant for ∑DEHP and MCPP. Among the LMWPs, race and ethnicity was a significant predictor of MiBP, MEP and MnBP; education was a significant predictor of MiBP, MEP and MnBP and clinical site of care was significant for MEP and MnBP. Lastly, for the replacement phthalates, education and clinical site of care were significant predictors of MECPTP concentrations and education and BMI was a significant predictor of MEHHTP.

## 4. Discussion

In this large sample of racially and ethnically diverse pregnant women across the US, we assessed their longitudinal PthM concentration levels throughout pregnancy using repeated urine samples, and then identified the predictors of PthMs. This study is one of few that has collected repeated measures of PthM concentrations using a nationally representative cohort consisting of nulliparous pregnant women with singleton pregnancies addressing previous limitations. We found significant differences by maternal age (i.e., ∑DEHP and MECPP), race and ethnicity (i.e., MBZP, ∑DEHP, MCPP, MECPP, MEHHP, MEHP, MEOHP, MnBP, MiBP, MECPTP), education (i.e., ∑DEHP, MCPP, MEOHP, MnBP, MEP, MEHHTP), pre-pregnancy BMI (i.e., MCOP, MCPP, MiBP) and clinical site of care (i.e., MBZP, ∑DEHP, MCOP, MCPP, MECPP, MEHHP, MEHP, MEOHP, MiNP, MnBP, MEP) in PthMs across pregnancy. In our study, the PthM concentration levels did not vary by trimester; however, DEHP followed by MEP, MiBP and MBZP had the highest detected concentrations throughout pregnancy. This is similar to a cross-sectional study of pregnant women in Charleston, South Carolina, where the concentrations of MEP were highest, followed by ∑DEHP, MiBP and MBzP [29]. Using NHANES 2002–2003 data, Woodruff et al. also found MEP to have the highest average concentrations among pregnant and non-pregnant woman in the US [30]. However, studies conducted in Asia found higher concentrations of MiBP and MEP in comparison to ∑DEHP [12,31]. While Western and European countries, including our study, had a high detection frequency for maternal urinary MBzP (>95% detection frequency) [32,33,34], MBzP was not commonly detected in Asian pregnant women (<50% detection rate) [12,31,35]. The replacement phthalate metabolites MEHHTP and MECPTP had detection frequencies >80% in our sample and this is similar to a longitudinal study among pregnant women in the PROTECT birth cohort in Puerto Rico [36]. But, MNHCH only had a detection rate of 22% in our sample. The differences in detection frequency, variations in exposure to product use and diet based on geography may highlight the differential concentration patterns of phthalates and socioeconomic factors.

In our sample, Non-Hispanic Black, Hispanic and Asian women had higher concentrations of MBZP, MnBP, MiBP and MECPTP throughout pregnancy in comparison to non-Hispanic White women. The findings are similar to a study by James-Todd et al. that determined whether PthM concentrations differed by race and ethnicity across multiple pregnancy time points in a sample that was composed of 16% African American, 14% Hispanic, 5% Asian and 59% white pregnant women. James-Todd et al. concluded that baseline levels of PthMs were significantly higher among non-White pregnant women and MEP and MCPP levels had the most significant changes across pregnancy [15]. In addition, patterns of phthalate concentrations varied across clinical sites throughout the US, underscoring the differences in consumer exposure to available products containing phthalates. The prevalence of specific personal care products use varies by race [16,37]. For example, certain hair products are more used by non-Hispanic Black women in comparison to women of other races, and previous studies have highlighted the differential concentration patterns of phthalates [3,4,38,39].

A recent study using cross-sectional data from 754 Black women from Detroit, Michigan, found that MEP concentrations were positively associated with personal care product use and specifically with nail products [40]. Another study of pregnant women found a prevalence of 45% and 41% for perfume use for Black and Hispanic pregnant women, respectively [17], whereas a different study that predominantly consisted of White pregnant women found a prevalence of 31% for perfume use [16]. Phthalates such as DEHP and DEP are typically found in perfumes. Braun et al. in 2013 found a relationship between self-reported personal care product use and urinary phthalate concentrations in pregnant women whereby increased use of lotions, cosmetics, perfumes, hair products, nail polish, etc. was linked with higher concentrations of urinary PthMs, specifically MEP and MBP [16]. The differential exposure to phthalates by race and ethnicity raises concerns about adverse maternal and fetal health outcomes among specific populations that have increased PthM levels during pregnancy [11]. More studies are needed to identify phthalate concentration levels based on product use in a racially diverse population to discern whether product use explains the differential exposures to phthalate metabolites in pregnant women. Few studies have explored the associations between sociodemographic predictors of exposure, specifically among populations at risk of high exposure levels.

Previously, associations between phthalate exposures and BMI have been noted, and our study parallels these previous findings. Specifically, among women exposed to high concentration levels (≥4 metabolites with concentration levels ≥90th percentile), having a BMI in the obese range was a strong predictor of having high PthM concentration levels. This may reflect the potential obesogenic actions of PthMs since PthMs are considered “obesogenic” agents or “metabolic disruptors” that can disrupt the body’s weight set point by interfering with processes related to glucose and lipid metabolism, insulin sensitivity and the regulation of sex steroids [41]. Phthalates have also been associated with maternal and childhood obesity [42,43,44,45,46,47], as a possible result of prenatal exposure. Studies from the US included a diverse pregnant population with larger proportions of overweight and obese women at baseline. Poor diet is associated with high phthalate exposure whereby phthalates can enter food during processing and packaging. Diet is an important source of exposure for most phthalates and replacement phthalates. People who dine out or consume large amounts of fast foods have greater exposure to heavily processed and packaged foods and generally have higher phthalate concentration levels than people who do not [48,49,50,51]. Women with more adipose tissue may have worse diets that may contribute to higher phthalate exposure levels and increased weight gain during pregnancy, highlighting another factor contributing to the inconsistency in the findings. While it is difficult to discern the causal direction of exposure to PthMs and BMI, this is a prediction study and our goal was to develop predictive models of phthalate exposure based on a set of observed predictors.

Contrary to other studies that found an association between low education attainment and increased phthalate concentrations [12,29,52,53], our study did not find such an association except for with MEP. In fact, for MEHHTP, the reverse association was found, whereby women who completed college and degree work beyond college where women higher education had higher levels of concentration in comparison to women with less than high school graduation. This may be because more educated women may use fewer products with traditional phthalates and use products that may contain more phthalate replacements, especially if replacement products are labeled “phthalate-free”. Wenzel et al., in a cross-sectional study using urine samples collected in the second trimester of pregnancy, found that Black women with less education (compared to college educated women), lower income and a higher BMI had an increased phthalate burden [29]. Another cross-sectional study in China found lower education to be associated with increased phthalate concentration [12]. In addition, increased levels of MBzP and MBP were associated with younger age, lower education and lower income in a cohort study of pregnant women in the Netherlands [52]. Similarly, lower education attainment was associated with higher levels of ∑DEHP and MEP in a repeated phthalate-measured Spanish cohort of pregnant women [53]. Perhaps this may call attention to the lack of awareness and education on phthalate contamination among pregnant women of lower socioeconomic positions. The variability of the findings in our study compared to others may be attributed to differences in population characteristics, sources of exposure and the number of and time of day at which urine samples were collected. Further, our sample is composed of pregnant women without adverse pregnancy complications including gestational diabetes, while other studies have associated higher phthalate concentrations with adverse pregnancy outcomes [6,7,10,11,54,55]. Thus, the patterns and concentration of PthMs may differ for women with adverse pregnancy events and other comorbid conditions, and our sample may not be representative of the exposure patterns of women with adverse pregnant outcomes. The lack of data on product use and dietary exposure is a limitation of our study as these factors are possible routes of exposure. Results from additional analysis, including identifying predictors among women with high concentration levels (≥4 metabolites with concentration levels ≥90th percentile) and identifying predictors over the geometric mean (over all trimesters) for each PthM, found similar associations and confirmed our findings from the main analysis.

Studies have reported the poor to moderate reliability of phthalate measurements between trimesters and high reliability for within-woman variability in phthalates [17], and our study confirms this finding. A US-based study by Yazdy et al., with predominantly Non-Hispanic White women, collected up to six urine samples during the second and third trimesters of pregnancy and found that the ∑DEHP concentration levels were not as reproducible (ICC: 0.32); however, MEP metabolite concentrations showed slightly higher reproducibility during pregnancy (ICC: 0.68) [56]. In our study, both the ∑DEHP (ICC = 0.03) and MEP (ICC = 0.11) levels were not reproducible over pregnancy. These findings highlight that multiple spot urine samples are required to ascertain exposure to phthalate compounds throughout pregnancy accurately.

Our study has several strengths. First, we measured PthM concentrations up to three times during pregnancy. Repeated urine samples provide a more accurate representation of the average concentration levels in the entire pregnancy period. Second, the representation of a diverse sample of pregnant women from a range of clinical sites enhances our findings’ generalizability to the US pregnant population. Third, the high percentage of observations above the LOD and the large sample size contribute to the robustness and internal validity of our findings. Lastly, we also examined newer replacement phthalate compounds since less is known about their effects on pregnant women.

A limitation that is worth noting is that PthMs are non-persistent chemicals and may be affected by variability within and across trimesters, and the exposure measure only captures exposure in the past day (or even less). While having multiple PthM measures may provide better exposure ascertainment throughout pregnancy, it may still be an inaccurate representation of long-term PthM exposure levels. In the NuMoM2b cohort, the PthM concentrations via urine samples were measured up to three times during pregnancy; however, because of their short biological half-lives and temporal variability during pregnancy, this measure may not fully characterize exposure levels throughout pregnancy.

## 5. Conclusions

Our study identified the prevalence and predictors of PthMs in a racially diverse population of pregnant women with singleton pregnancies throughout the US. Phthalate concentrations varied based on maternal sociodemographic characteristics, specifically race and ethnicity, pre-pregnancy BMI and clinical sites of care. Our findings suggest differential concentration patterns among heterogenous groups of women and highlight the need to address the risks for specific groups to reduce the burden of phthalate exposure as a means of preventing adverse health outcomes for women and their offspring. Accordingly, the American College of Obstetricians and Gynecologists (ACOG) has recommended screening pregnant women for environmental chemicals before and during pregnancy with counseling on how to reduce exposure (e.g., use fragrance-free rather than scented or unscented products) and the associated reproductive and developmental risks [56].

## Figures and Tables

**Table 1 ijerph-20-07104-t001:** Phthalate and replacement phthalate detection rates from the nuMoM2b nested case–control study (*n* = 953).

Molecular Weight	Parent Phthalate	Abbreviation	Metabolites	Abbreviation	LOD (ng/mL)	Percent Detect (%)
High	Butyl benzyl phthalate	BBzP	Monobenzyl	MBZP	0.2	99
	Di-(2-ethylhexyl) phthalate	DEHP	Mono(2-ethyl)-hexyl	MEHP	1	73
			Mono(2-ethyl-5-oxohexyl)	MEOHP	0.1	100
			Mono(2-ethyl-5-hydroxyhexyl)	MEHHP	0.1	100
			Mono2-ethyl-5-carboxypentyl	MECPP	0.2	100
			Monocarboxyoctyl	MCOP	0.2	96
			Monocarboxyisooctyl ester	MCOCH	0.2	5
	D-*n*-octyl phthalate	DnOP	Mono(3-carboxypropyl)	MCPP	0.2	97
	Di-2-ethylhexyl terephthalate	DEHTP *	Mono-2-ethyl-5-carboxypentyl terephthalate	MECPTP *	0.2	98
	Di-2-ethylhexyl terephthalate		Mono-2-ethyl-5-hydroxyhexyl terephthalate	MEHHTP *	0.2	81
	Di(isononyl) cyclohexane-1,2-dicarboxylate	DINCH *	Mono(cis-hydroxy-isononyl) ester	MHNCH *	0.2	22
	Di-isononyl phthalate	DiNP	Mono-isononyl	MINP	0.5	13
			Monocarboxy Isononyl Phthalate	MCINP	0.3	66
Low	Di-isobutyl phthalate	DiBP	Monoisobutyl	MiBP	0.1	100
	Diethyl phthalate	DEP	Monoethyl	MEP	1.1	100
	Di-*n*/i-butyl phthalate	DnBP	Monobutyl	MNBP	0.5	100

LOD = Limit of detection; * Replacement parent compound and metabolites.

**Table 2 ijerph-20-07104-t002:** Demographic characteristics of controls within the Nulliparous Mothers To Be (nuMoM2b) nested case–control study (*n* = 953) enrolled between 2010–2015.

Maternal Characteristics	No.	%
**Age (years)**		
**<20**	119	12.5
**20–34**	735	77.1
**≥35**	99	10.4
**Maternal race and ethnicity**		
**Non-Hispanic White**	552	57.9
**Non-Hispanic Black/African American**	138	14.5
**Hispanic**	172	18.1
**Asian**	53	5.6
**American Indian, Native Hawaiian, other** **or more than one race**	38	4.0
**Education level**		
**Less than HS grad**	98	10.3
**HS grad or GED**	101	10.6
**Some college or assoc/tech degree**	263	27.6
**Completed college**	266	27.9
**Degree work beyond college**	225	23.6
**Pre-pregnancy BMI (kg/m^2^)**		
**Underweight < 18.5**	47	5.9
**Normal 18.5–<25**	494	51.8
**Overweight 25–30**	233	24.5
**Obese > 30**	179	18.8
**Clinical Centers**		
**North East Region**		
**Columbia University, NY**	210	22.0
**Magee-Women’s Hospital, PA**	130	13.6
**University of Pennsylvania, PA**	70	7.4
**Mountain States**		
**University of Utah, UT**	189	19.8
**Midwest Region**		
**Case Western Reserve University, OH**	63	6.6
**Indiana University, IN**	88	9.2
**Northwestern University, IL**	132	13.9
**Western Region**		
**University of California Irvine, CA**	71	7.5

**Table 3 ijerph-20-07104-t003:** Distribution of specific-gravity-adjusted trimester specific urinary phthalate metabolites measured during pregnancy.

Parent Phthalate	Phthalate Metabolites (ng/mL)	Trimester	Arithmetic Mean	Geometric Mean	Arithmetic Percentile				
					Min	25th	50th	75th	Max
**BBzp**	MBzP	1	10.1	5.3	0.1	2.6	5.0	10.3	354.0
		2	10.6	5.6	0.1	2.6	5.1	10.7	151.9
		3	11.7	5.4	0.2	2.6	5.0	9.8	440.8
**DEP**	MEP	1	203.9	47.9	0.6	18.3	40.7	109.7	40,904.0
		2	204.1	60.0	0.5	17.6	41.4	111.6	6560.2
		3	209.5	50.8	0.3	0.8	18.1	42.6	115.3
**DiBP**	MiBP	1	14.0	9.5	0.1	5.6	9.1	15.0	695.3
		2	15.4	10.1	0.1	4.9	10.7	23.2	1552.2
		3	14.3	9.9	0.1	5.6	9.1	15.0	695.3
**DEHP**	MEHP, MEOHP, MEHHP, MECPP	1	290.8	146.4	0.7	88.5	133.0	216.2	6516.1
		2	225.8	138.8	0.1	84.4	130.3	207.6	6516.1
		3	242.7	141.9	0.5	84.9	134.1	213.6	18,284.0
**DEHP**	**MECPP**	1	22.3	12.0	0.1	7.1	11.0	19.0	2683.0
		2	18.2	11.5	0.1	6.6	10.9	17.5	409.9
		3	19.7	12.0	0.2	6.9	11.2	18.2	1167.7
**DEHP**	**MEHHP**	1	47.4	21.6	0.1	12.7	20.5	33.5	8198.2
		2	34.8	20.0	0.0	12.1	19.3	32.3	1117.9
		3	36.6	20.0	0.1	11.5	19.6	32.3	2616.1
**DEHP**	MEHP	1	4.9	2.6	0.4	1.5	2.4	4.1	371.5
		2	4.4	2.7	0.4	1.6	2.4	4.1	141.9
		3	5.1	2.7	0.5	1.5	2.4	4.2	729.6
**DEHP**	**MEOHP**	1	11.7	5.9	0.1	3.6	5.3	8.6	1662.6
		2	9.7	5.9	0.0	3.5	5.6	8.8	283.8
		3	10.7	6.2	0.1	3.7	5.8	9.0	867.2
**DiNP**	**MCOP** **MiNP**	1	5.9	2.6	0.1	1.0	2.4	6.2	102.5
		2	5.2	2.3	0.1	0.9	2.0	5.3	108.4
		3	6.1	2.6	0.2	1.0	2.3	6.0	160.0
		1	0.7	0.5	0.2	0.3	0.4	0.8	9.0
		2	0.8	0.5	0.1	0.3	0.4	0.8	18.8
		3	1.0	0.5	0.2	0.3	0.4	0.8	75.7
**DnOP MCPP**		1	12.1	2.7	0.1	1.2	2.2	4.7	2874.7
		2	9.3	2.6	0.1	1.2	2.1	4.5	831.3
		3	9.8	2.6	0.2	1.2	2.1	5.5	1358.7
**DEHTP * MECPTP *** **MEHHTP ***		1	17.6	3.5	0.1	1.2	2.8	6.8	1742.0
		2	19.2	3.9	0.1	1.4	3.2	8.5	853.3
		3	21.9	4.6	0.2	1.6	3.5	9.2	1113.8
		1	2.4	0.8	0.1	0.4	0.6	1.3	183.3
		2	2.3	0.8	0.1	0.4	0.6	1.3	97.08
		3	2.3	0.8	0.1	0.4	0.6	1.3	110.6
**DnBP MNBP**		1	19.4	13.1	0.3	8.0	12.6	20.3	1831.7
		2	20.1	13.9	0.2	8.5	13.4	21.7	617.7
		3	18.4	13.5	0.4	8.5	12.7	21.0	503.8

ΣDEHP=(MEHP/278)+(MEOHP/292)+(MEHHP/294)+(MECPP/308), in nmol/L [13]. * Replacement compound and metabolites.

**Table 4 ijerph-20-07104-t004:** Intra-class correlation coefficients (ICCs) over the three measures for urinary phthalate metabolite concentrations (*n* = 2723 urine samples).

Phthalate Metabolite	ICC
**MBZP**	0.52
**DEHP**	0.03
**MCOP**	0.11
**MCPP**	0.001
**MECPP**	0.04
**MEHHP**	0.02
**MEHP**	0.07
**MEOHP**	0.04
**MINP**	0.02
**MNBP**	0.10
**MEP**	0.11
**MIBP**	0.53

**Table 5 ijerph-20-07104-t005:** GEE population-averaged model for predictors of urinary concentration of *high-molecular-weight* phthalate (*HMWP*) metabolites (ng/mL) (*n* = 953).

Predictors	HMWP Metabolites								
	MBZP	∑DEHP	MCOP	MCPP	MECPP	MEHHP	MEHP	MEOHP	MiNP
**Adjusted Model**	Estimate (95% Cl)								
**Trimester**									
**1**	REF								
**2**	0.87 (−0.17, 1.92)	−65.90 (−168.26, 36.46)	−0.37 (−1.18, 0.43)	−2.97 (−10.04, 4.10)	−4.12 (−10.68, 2.43)	−12.81 (−31.69, 6.08)	−0.46 (1.77, 4.86)	−2.10 (−6.08,1.89)	0.06 (−0.05, 0.16)
**3**	1.67 (−0.08, 3.42)	−47.63 (−154.74, 59.48)	0.50 (−0.42, 1.42)	−2.35 (−10.09, 5.39)	−2.55 (9.34, 4.23)	−10.84 (−30.19, 8.52)	0.22 (−1.65, 0.72)	−0.97 (−5.24,3.30)	0.24 (0.02, 0.46)
***p*-value**	0.10	0.41	0.1	0.71	0.39	0.40	0.63	0.453	0.081
**Age (years)**									
**<20**	REF								
**20–34**	−0.67 (−5.59, 4.26)	38.32 (−20.46, 97.11)	0.57 (−0.92, 2.05)	−1.79 (−6.86, 3.28)	3.20 (−1.28,7.67)	6.41 (−3.70,16.53)	0.43 (−0.79, 1.66)	1.34 (−1.14, 3.82)	0.06 (−0.11, 0.22)
**≥35**	−3.46 (−9.21, 2.29)	−32.72 (−116.68, 51.23)	0.19 (−1.58, 1.96)	9.22 (−10.48, 28.93)	−2.30 (−8.29, 3.68)	−4.53 (−18.14, 9.09)	−1.28 (−3.72, 1.17)	−1.48 (−5.21, 2.26)	0.03 (−0.33, 0.38)
***p*-value**	0.23	0.04	0.60	0.57	0.02	0.07	0.11	0.08	0.78
**Race/ethnicity**									
**Non-Hispanic White**	REF								
**Non-Hispanic Black/African American**	3.71 (−0.97, 8.39)	−70.19 (−166.69, 26.32)	0.17 (−1.17, 1.52)	−5.90 (−11.89, 0.09)	−6.17 (−12.28, −0.06)	−11.32 (−29.09, 6.45)	−0.39 (−1.71, 0.94)	−3.09 (−6.91, 0.73)	−0.12 (−0.33, 0.09)
**Hispanic**	1.35 (−1.38, 4.08)	−27.16 (−129.59, 75.28)	0.48 (−0.94, 1.89)	−5.92 (−13.42, 1.57)	−1.05 (−7.83, 5.72)	−5.52 (−23.56, 12.52)	−0.11 (−2.15, 1.94)	−1.31 (−5.53, 2.91)	−0.01 (−0.18, 0.17)
**Asian**	3.79 (−2.58, 10.16)	−43.55 (−138.00, 50.90)	0.14 (−2.23, 2.51)	−7.74 (−13.94, −1.53)	−0.10 (−8.34, 8.14)	−9.38 (−23.14, 4.38)	−1.15 (−3.96, 1.67)	−2.13 (−6.30, 2.04)	−0.04 (−0.25, 0.17)
**American Indian, Native Hawaiian, other or more than 1 race**	−3.10 (−5.89, −0.32)	−112.90 (−195.92, −29.88)	0.19 (−1.59, 1.97)	−8.95 (−14.22, −3.67)	−7.94 (−13.27, −2.61)	−18.66 (−33.39, −3.93)	−1.70 (−3.52, 0.13)	−5.14 (−8.41, −1.87)	−0.08 (−0.35, 0.19)
***p*-value**	0.355	0.070	0.600	0.006	0.01	0.05	0.24	0.01	0.80
**Education level**									
**Less than HS grad**	REF								
**HS grad or GED**	0.40 (−3.97, 4.77)	−65.78 (−125.82, −5.73)	−0.70 (−2.59, 1.19)	−2.06 (−6.92, 2.81)	−5.04 (−10.16, 0.07)	−10.40 (−20.24, −0.57)	−1.02 (−2.46, 0.43)	−3.02 (−5.72, −0.32)	−0.06 (−0.27, 0.15)
**Some college or assoc/tech degree**	−0.21 (−5.04, 4.61)	40.05 (−44.67, 124.76)	0.41 (−1.40, 2.22)	1.67 (−3.47, 6.81)	1.76 (−4.54, 8.06)	8.67 (−6.06, 23.40)	0.34 (−1.36, 2.04)	1.13 (−2.41, 4.67)	0.06 (−0.15, 0.27)
**Completed college**	−0.41 (−5.37, 4.56)	−70.75 (−168.10, 26.60)	0.02 (−1.92, 1.96)	7.00 (−1.44, 15.45)	−6.01 (−13.16, 1.14)	−11.18 (−28.17, 5.82)	−0.44 (2.36,1.48)	−3.28 (−7.34, 0.77)	0.17 (−0.09, 0.43)
**Degree work beyond college**	−2.62 (−7.40, 2.15)	−3.11 (−116.16, 109.93)	0.04 (−2.02, 2.10)	7.07 (−2.09, 16.23)	−2.09 (−10.18, 6.00)	0.15 (−17.94, 18.24)	1.20 (−2.22, 4.63)	−0.21 (−5.25, 4.82)	0.12 (−0.14, 0.37)
***p*-value**	0.36	0.07	0.60	0.01	0.16	0.06	0.10	0.07	0.13
**Pre-pregnancy BMI**									
**Normal 18.5–<25**	REF								
**Underweight <18.5**	−0.06 (−4.08, 3.97)	−8.11 (−81.45, 65.22)	0.61 (−1.31, 2.54)	5.33 (3.43, 14.10)	0.16 (−6.16, 6.48)	−3.28 (−14.69, 8.13)	0.09 (−2.20, 2.38)	0.58 (−3.10, 4.26)	−0.02 (−0.21, 0.16)
**Overweight 25–<30**	−0.26 (−2.79, 2.28)	17.14 (−39.37, 73.64)	1.72 (0.69, 2.75)	8.00 (−0.07, 16.08)	1.63 (−2.49, 5.75)	3.04 (−6.18, 12.26)	−0.32 (−1.78, 1.14)	0.76 (−1.71, 3.23)	0.02 (−0.12, 0.16)
**Obese ≥30**	4.14 (−0.06, 8.33)	109.63 (−68.77, 288.04)	1.57 (0.35, 2.80)	5.40 (−0.51, 11.30)	6.62 (−4.40, 17.65)	20.78 (−12.51,54.08)	0.91 (−1.13, 2.96)	4.13 (−2.74, 11.00)	0.12 (−0.25, 0.50)
***p*-value**	0.26	0.64	0.00	0.07	0.66	0.53	0.59	0.70	0.88
**Clinical site**									
**Case Western Reserve University, OH**									
**Columbia University, NY**	−11.28 (−22.67, 0.12)	110.62 (22.75,198.49)	1.83 (0.24, 3.43)	1.99 (−11.85, 15.84)	8.50 (1.88, 15.11)	16.77 (3.40, 30.14)	2.67 (−0.12, 5.45)	4.70 (0.72, 8.67)	0.20 (0.05, 0.34)
**Indiana University, IN**	−9.65 (−21.19, 1.88)	12.25 (−49.55, 74.05)	1.33 (−0.23, 2.90)	3.20 (−10.49, 16.88)	1.04 (−4.17, 6.24)	0.94 (−9.38, 11.26)	0.62 (−0.45, 1.68)	0.96 (−1.68, 3.60)	0.11 (−0.07, 0.29)
**Magee-Women’s Hospital, PA**	−12.69 (−24.26, −1.08)	−8.02 (−68.95, 52.92)	1.30 (−0.24, 2.84)	−5.78 (−15.13, 3.58)	−0.94 (−5.71, 3.84)	−1.38 (−11.52, 8.77)	0.26 (−1.10, 1.61)	−0.29 (−2.87, 2.28)	0.21 (0.05, 0.38)
**Northwestern University, IL**	−13.34 (−25.04, −1.63)	37.18 (−40.79, 115.15)	1.25 (−0.32, 2.81)	−4.76 (−17.10, 7.57)	2.01 (−3.80, 7.82)	6.46 (−6.27, 19.20)	1.08 (−1.11, 3.28)	1.35 (−1.90, 4.61)	0.43 (−0.04, 0.90)
**University of California Irvine, CA**	−16.52 (−27.87, −5.18)	−26.39 (−86.84, 34.06)	−1.03 (−2.63, 0.57)	−6.07 (−16.21, 4.07)	−3.91 (−8.88, 1.06)	−2.70 (−12.86, 7.46)	−0.39 (−1.44, 0.66)	−0.92 (−3.54, 1.70)	−0.06 (−0.20, 0.08)
**University of Pennsylvania, PA**	−10.65 (−23.42, 2.12)	133.61 (16.97, 250.26)	2.39 (0.57, 4.21)	3.71 (−7.96, 15.39)	8.12 (0.06, 16.18)	23.10 (3.29, 42.92)	2.59 (0.04, 5.13)	5.57 (0.69, 10.46)	0.37 (0.03, 0.72)
**University of Utah, UT**	−12.87 (−24.05, −1.69)	129.15 (−4.98, 263.29)	2.48 (0.79, 4.17)	−7.47 (−17.33, 2.39)	8.15 (−0.89, 17.19)	23.11 (−1.36, 47.59)	1.60 (−0.04, 3.23)	5.38 (0.10, 10.65)	0.40 (0.20, 0.60)
***p*-value**	0.00	0.00	0.00	0.03	0.00	0.01	0.00	0.00	0.00

**Table 6 ijerph-20-07104-t006:** GEE population-averaged model for predictors of urinary concentration of *low-molecular-weight* phthalate (*LMWP*) metabolites (ng/mL) (*n* = 953).

Predictors	LMWP Metabolites		
	**MnBP**	**MEP**	**MiBP**
**Adjusted Model**			
**Trimester**			
**1**	REF		
**2**	0.80 (−3.60, 5.21)	0.50 (−109.81, 110.81)	1.55 (−0.37, 3.47)
**3**	−0.83 (−5.02, 3.37)	4.93 (−96.28, 106.14)	0.46 (−1.32, 2.25)
***p*-value**	0.35	0.99	0.21
**Age (years)**			
**<20**			
**20–34**	5.13 (−0.12, 10.37)	120.56 (−24.67, 265.79)	1.03 (−1.97, 4.03)
**≥35**	4.88 (−2.28, 12.03)	52.22 (−97.55, 201.99)	0.38 (−3.43, 4.18)
***p*-value**	0.16	0.09	0.77
**Race/ethnicity**			
**Non-Hispanic White**	REF		
**Non-Hispanic Black/African American**	2.39 (−3.85, 8.64)	105.52 (−51.23, 262.27)	4.96 (0.50, 9.43)
**Hispanic**	5.74 (1.13, 10.34)	6.54 (−102.95, 116.04)	6.06 (1.23, 10.89)
**Asian**	3.67 (−7.45, 14.78)	163.92 (−42.94, 370.78)	7.70 (−3.02, 18.42)
**American Indian, Native Hawaiian, other** **or more than 1 race**	−1.90 (−7.35, 3.55)	76.63 (−75.42, 228.68)	6.24 (−7.87, 20.35)
***p*-value**	0.06	0.33	0.00
**Education level**			
**Less than HS grad**			
**HS grad or GED**	−6.84 (−13.55, −0.13)	21.74 (−222.18, 265.66)	−1.70 (−5.00, 1.60)
**Some college or assoc/tech degree**	−5.71 (−12.85, 1.43)	−61.06 (−181.58, 59.45)	1.37 (−2.77, 5.52)
**Completed college**	−12.12 (−19.39, 4.85)	−87.87 (−239.33, 63.59)	−2.36 (−6.21, 1.49)
**Degree work beyond college**	−9.72 (−18.90, −0.53)	−187.99 (−332.61, −43.37)	−1.75 (−7.72, 4.22)
***p*-value**	0.00	0.03	0.51
**BMI (at visit 1)**			
**Normal 18.5–<25**	REF		
**Underweight <18.5**	−5.21 (−9.65, −0.76)	−25.31 (−197.49, 146.86)	−3.00 (−5.88, −0.12)
**Overweight 25–<30**	−0.56 (−3.73, 2.61)	83.39 (−61.31, 228.10)	0.21 (−3.61, 4.03)
**Obese ≥30**	−0.12 (−3.34, 3.10)	−16.67 (−102.39, 69.05)	2.13 (−0.72, 4.97)
***p*-value**	0.09	0.68	0.02
**Clinical site**			
**Case Western Reserve University, OH**	REF		
**Columbia University, NY**	−0.97 (−6.30, 4.35)	159.49 (19.86, 299.12)	4.32 (−1.14, 9.77)
**Indiana University, IN**	0.54 (−6.21, 7.29)	15.95 (−77.78, 109.67)	0.82 (−3.60, 5.23)
**Magee-Women’s Hospital, PA**	−4.30 (−9.58, 0.98)	55.48 (−39.66, 150.61)	−0.91 (−4.38, 2.56)
**Northwestern University, IL**	−5.98 (−11.76, −0.20)	150.70 (15.38, 286.02)	−0.77 (−4.67, 3.12)
**University of California Irvine, CA**	−8.59 (−14.31, −2.88)	300.57 (95.44, 505.70)	−4.99 (−10.08, 0.10)
**University of Pennsylvania, PA**	10.91 (−7.04, 28.87)	224.47 (−203.37, 652.32)	3.33 (−5.04, 11.70)
**University of Utah, UT**	−3.46 (−9.15, 2.23)	21.99 (−82.24, 126.22)	−0.26 (−3.98, 3.46)
***p*-value**	0.00	0.01	0.09

**Table 7 ijerph-20-07104-t007:** GEE population-averaged model for predictors of urinary concentration of *replacement* phthalate metabolites (ng/mL) (*n* = 953).

Predictors	Replacement Phthalate Metabolites	
	**MECPTP**	**MEHHTP**
**Adjusted Model**		
**Trimester**		
**1**	REF	
**2**	1.46 (−5.49, 8.41)	−0.14 (−0.94, 0.67)
**3**	4.22 (−3.34, 11.78)	−0.18 (−0.96, 0.61)
***p*-value**	0.54	0.91
**Age (years)**		
**<20**		
**20–34**	−1.35 (−14.02, 11.32)	−0.13 (−1.30,1.03)
**≥35**	1.18 (−17.93, 20.30)	0.26 (−1.73, 2.25)
***p*-value**	0.90	0.84
**Race/ethnicity**		
**Non-Hispanic White**	REF	
**Non-Hispanic Black/African American**	12.02 (2.81, 21.24)	0.66 (−0.16, 1.47)
**Hispanic**	11.77 (−3.49, 27.02)	1.17 (−0.41, 2.75)
**Asian**	4.16 (−7.91, 16.23)	0.04 (−1.17, 1.25)
**American Indian, Native Hawaiian, other** **or more than 1 race**	−4.73 (−11.60, 2.14)	−0.57 (−1.40, 0.25)
***p*-value**	0.01	0.07
**Education level**		
**Less than HS grad**	REF	
**HS grad or GED**	0.05 (−9.50, 9.59)	−0.13 (−0.98, 0.71)
**Some college or assoc/tech degree**	9.24 (−3.02, 21.51)	1.11 (−0.01, 2.23)
**Completed college**	11.41 (−0.93, 23.74)	1.24 (0.06, 2.43)
**Degree work beyond college**	9.32 (−2.96, 21.60)	1.28 (0.04, 2.51)
***p*-value**	0.17	0.01
**Pre-pregnancy BMI**		
**Normal 18.5–<25**	REF	
**Underweight <18.5**	−0.35 (−10.24, 9.54)	0.07 (−0.97, 1.10)
**Overweight 25–<30**	−4.03 (−11.84, 3.77)	−0.40 (−1.20, 0.41)
**Obese ≥30**	2.11 (−6.84, 11.07)	0.59 (−0.40,1.57)
***p*-value**	0.63	0.32
**Clinical site**		
**Case Western Reserve University, OH**	REF	
**Columbia University, NY**	−7.05 (−18.15, 4.06)	−0.60 (−1.60, 0.41)
**Indiana University, IN**	7.95 (−4.08, 19.99)	0.91 (−0.38, 2.19)
**Magee-Women’s Hospital, PA**	9.34 (−1.63, 20.32)	0.73 (−0.30, 1.76)
**Northwestern University, IL**	2.47 (−9.86, 14.80)	0.06 (−1.11, 1.24)
**University of California Irvine, CA**	−6.93 (−21.47, 7.61)	−0.96 (−2.19, 0.27)
**University of Pennsylvania, PA**	9.60 (−8.32, 27.53)	1.24 (−0.57, 3.05)
**University of Utah, UT**	15.63 (3.66, 27.59)	1.71 (0.54, 2.88)
***p*-value**	0.00	0.00

**Table 8 ijerph-20-07104-t008:** Predictors of phthalate metabolites for women exposed to ≥4 metabolites versus <4 metabolites among women with concentration levels ≥90th percentile.

Predictors	Women Exposed to ≥4 Metabolites vs. <4 Metabolites
**Adjusted Model**	Odds Ratio (95% Cl)
**Age (years)**	
**<20**	REF
**20–34**	1.15 (0.66, 2.01)
**≥35**	0.85 (0.41, 1.76)
***p*-value**	0.4152
**Race/ethnicity**	
**Non-Hispanic White**	REF
**Non-Hispanic Black/African American**	1.45 (0.89, 2.37)
**Hispanic**	1.28 (0.82, 2.00)
**Asian**	0.83 (0.44, 1.57)
**American Indian, Native Hawaiian, other** **or more than 1 race**	0.77 (0.36, 1.62)
***p*-value**	0.36
**Education level**	
**Less than HS grad**	REF
**HS grad or GED**	0.95 (0.49,1.84)
**Some college or assoc/tech degree**	1.28 (0.69, 2.38)
**Completed college**	1.17 (0.59, 2.30)
**Degree work beyond college**	1.46 (0.72, 2.98)
***p*-value**	0.59
**Pre-pregnancy BMI**	
**Normal 18.5–<25**	REF
**Underweight <18.5**	1.66 (0.81, 3.40)
**Overweight 25–<30**	1.30 (0.92, 1.85)
**Obese ≥30**	1.95 (1.31, 2.90)
***p*-value**	0.01
**Clinical site**	
**Case Western Reserve University, OH**	REF
**Columbia University, NY**	1.83 (0.93, 3.60)
**Indiana University, IN**	2.18 (1.03, 4.59)
**Magee-Women’s Hospital, PA**	1.33 (0.66, 2.69)
**Northwestern University, IL**	1.71 (0.81, 3.59)
**University of California Irvine, CA**	0.87 (0.38, 1.99)
**University of Pennsylvania, PA**	2.67 (1.22, 5.82)
**University of Utah, UT**	2.60 (1.30, 5.20)
***p*-value**	0.01

## Data Availability

Data available upon request.

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
