# Peer review of "Predictors of Phthalate Metabolites Exposure among Healthy Pregnant Women in the United States, 2010–2015"

_ijerph, 2023, doi:10.3390/ijerph20237104_

Round 1

Reviewer 1 Report

Comments and Suggestions for Authors

The manuscript, entitled “Predictors of Phthalate Metabolites Exposure Among Healthy Pregnant Women in The United States, 2010-2015.” examined determinants of phthalate metabolites in a cohort of racially/ethnically diverse nulliparous pregnant women. The findings indicate exposure patterns that require policies to reduce exposure in specific subgroups. However, due to some drawbacks, my suggestion is rejection.

Comments:

1. The article has serious flaws, and additional data are needed. The title of this article is Predictors of Phthalate Metabolites Exposure Among Healthy Pregnant Women…, but in the text the authors did not the effect of phthalate metabolites on pregnant woman and fetus. The research significance of this paper is not clear.

2. The information before and after the manuscript is inconsistent. In abstract section, it is clarified that nine parent phthalates were reported including Butyl benzyl phthalate (BBzP), Diisobutyl phthalate (DiBP), Diethyl phthalate (DEP), Diisononyl phthalate (DiNP), D-n-octyl phthalate (DnOP), Di-2-ethylhexyl terephthalate (DEHTP), Di-n/i-butyl phthalate (DnBP), Di-isononyl phthalate (DiNP) and Di-(2-ethylhexyl) phthalate (DEHP) from urine. But in table 2, 10 parent phthalate were shown. What’s more, the sample size is 953 in table 1, however the sample size is 960 in table 2.

3. “PthMs are non-persistent compounds with short half-lives, that mostly reflect exposure in the past 3 to 24 hours”. If PthMs can exist in urine persistently? Do the short half-lives affect the detection results?

4. in introduction section, the importance and meaning of this study are suggested to be supplemented.

5. The strength and weakness of this study should also be added in discussion section.

6. The maternal race and ethnicity were classified into Non-Hispanic White, Non-Hispanic Black/African American, Hispanic…. What are the racial classification criteria in the article?

7. The formatting of tables should keep consistent. For example, some words in table 3 are in bold, but others are not.

Author Response

Please find attached response to reviewer 1

Reviewer 2 Report

Comments and Suggestions for Authors

This study is interesting, but it was not always easy and straight-forward to understand. In particular, the way results are presented. Too many tables and data. However, I understand that all the data can hardky be represented in a different format. 

If I have understood correctly, the samples were collected between 2010 and 2015. 

Limitations in study methodology, may be not having used a pool of urine collected at different times of the day, because there are day-to-day variations of phthalate urinary levels, also different in individual subjects.

Minor points: 

Line 131-134: DINCH is not cited among the nine parent phthalates

I cannot find the Title and description of Table 6.

Author Response

Please find attached response to reviewer 2

Reviewer 3 Report

Comments and Suggestions for Authors

No comments about the presentation, data, statistical analysis. The only comment would be regarding this compounds, I would recommend some more general information about them, and some idea in the conclusion  - what should pregnant women to avoid

Author Response

Please find attached response to reviewer 3
